# A Comparison of Functional Connectivity in the Human Brainstem and Spinal Cord Associated with Noxious and Innocuous Thermal Stimulation Identified by Means of Functional MRI

**DOI:** 10.3390/brainsci13050777

**Published:** 2023-05-09

**Authors:** Elena Koning, Jocelyn M. Powers, Gabriela Ioachim, Patrick W. Stroman

**Affiliations:** 1Centre for Neuroscience Studies, Queen’s University, Kingston, ON K7L 3N6, Canada; 2Department of Biomedical and Molecular Sciences, Queen’s University, Kingston, ON K7L 3N6, Canada; 3Department of Physics, Queen’s University, Kingston, ON K7L 3N6, Canada

**Keywords:** spinal fMRI, somatosensation, descending modulation of pain, pain sensitivity, neural connectivity

## Abstract

The somatosensory system is multidimensional and processes important information for survival, including the experience of pain. The brainstem and spinal cord serve pivotal roles in both transmitting and modulating pain signals from the periphery; although, they are studied less frequently with neuroimaging when compared to the brain. In addition, imaging studies of pain often lack a sensory control condition, failing to differentiate the neural processes associated with pain versus innocuous sensations. The purpose of this study was to investigate neural connectivity between key regions involved in descending modulation of pain in response to a hot, noxious stimulus as compared to a warm, innocuous stimulus. This was achieved with functional magnetic resonance imaging (fMRI) of the brainstem and spinal cord in 20 healthy men and women. Functional connectivity was observed to vary between specific regions across painful and innocuous conditions. However, the same variations were not observed in the period of anticipation prior to the onset of stimulation. Specific connections varied with individual pain scores only during the noxious stimulation condition, indicating a significant role of individual differences in the experience of pain which are distinct from that of innocuous sensation. The results also illustrate significant differences in descending modulation before and during stimulation in both conditions. These findings contribute to a deeper understanding of the mechanisms underlying pain processing at the level of the brainstem and spinal cord, and how pain is modulated.

## 1. Introduction

Pain is a complex and highly conserved phenomenon in humans, providing useful information about our environment in order to avoid potential or actual harm. Although neuroimaging studies of thermal pain have proven useful for investigating pain mechanisms in healthy and diseased states, most studies have been conducted in the brain. The spinal cord and brainstem serve essential roles in the processing of information related to pain and other somatosensory stimuli. The development of advanced neuroimaging techniques has vastly accelerated our understanding of network connectivity in the brainstem and spinal cord. In particular, functional magnetic resonance imaging (fMRI) studies have attributed essential roles of pain modulation to the periaqueductal gray matter (PAG), rostroventromedial medulla (RVM), and other brainstem regions involved in regulating pain through descending pathways [1,2,3,4]. Significant effects of emotional–motivational, cognitive, and attentional factors on perception have also been demonstrated, including the effects of anticipation, an emotional state which has been shown to alter activity in the brainstem and spinal cord regions when expecting a painful stimulus [5]. In addition, previous work by our group has shown that pain responses vary significantly across individuals, adding to other research describing the highly subjective and personal nature of the pain experience [6,7,8].

Previous studies using fMRI in the human brainstem and spinal cord have demonstrated neural signalling and connectivity along the PAG-RVM-cord pathway that is attributed to descending pain regulation, as well as input to the spinal cord from the dorsal reticular nucleus of the medulla (DRt) [6]. The results demonstrated that the regulation of spinal cord function is a continuous process that varies in relation to a person’s emotional and cognitive state, such as expecting a noxious stimulus, or knowing that the stimulus has ended. A following study provided more details about this continuous regulation and showed variations in connectivity between brainstem regions that are correlated with pain ratings in healthy participants, not only during a noxious stimulus, but before and after stimulation as well [9]. These results demonstrate the dynamic nature of descending pain regulation and how pain sensitivity is regulated in relation to a person’s emotional and cognitive state. However, these findings raise the question of whether it is possible to identify neural signalling in the brainstem and spinal cord that is specific to the experience of pain, and does not occur in the absence of pain.

In the present study, we therefore aimed to identify features of neural signalling in the brainstem and spinal cord that are specific to the experience of pain. The objective was to use our established methods for fMRI of the brainstem and spinal cord to differentiate neural activity and functional connectivity associated with two similar sensory stimuli; a noxious hot stimulus and an innocuous warm thermal stimulus. We hypothesized that the results would demonstrate significant differences in functional connectivity between noxious and innocuous thermal stimuli across pre-determined networks associated with pain processing, as well as a significant influence of individual differences on pain perception in healthy individuals.

## 2. Materials and Methods

### 2.1. Participant Recruitment

All study procedures were reviewed and approved by our institutional human research ethics board. Additional safety precautions such as reduced study personnel and personal protective equipment were implemented as data collection occurred during the COVID-19 pandemic. Participants were recruited from the local community through online advertisements and were screened to ensure that they were free of any contraindications for the MRI environment, including claustrophobia, metal implants, a history of injury from metal, or pregnancy. Other exclusion criteria included the ongoing use of centrally acting medications, history of neurological disease or injury, psychiatric illness, or the presence of a pre-existing pain condition or other major medical illness.

A total of 10 healthy women and 10 healthy men were recruited. This study was part of a larger project which included separate brain and brainstem/spinal cord imaging sessions; the research contained in this paper concerns the brainstem/spinal cord data. All participants provided informed consent prior to participating and were informed that they could terminate their involvement in the study at any time. Participants completed a series of questionnaires including the State-Trait Anxiety Inventory, Social Desirability Scale, Beck Depression Inventory, Pain Catastrophizing Scale, and a form to collect demographic information [10,11,12,13]. Questionnaire results were not analyzed for the purposes of the current investigation and will be used for future studies investigating mental health status, pain symptoms, and functional MRI results.

### 2.2. Sham MRI Training

Prior to data collection, participants were trained in a “sham” MRI lab within the Queen’s University MRI Facility in order to familiarize them with the stimulus, pain rating scales, study procedures, and the confined space of a mock-up of the MRI. Researchers also answered any questions or concerns that participants expressed, and helped to alleviate any anxiety related to noxious stimulation or participating in the neuroimaging portion of the study. This study employed a thermal stimulation paradigm, applied to the skin overlying the thenar eminence on the right hand (corresponding to the C6 dermatome). Heat stimuli were applied using a custom-made MRI-compatible robotic contact-heat thermal stimulator (RTS-2), which consists of a pneumatically controlled, retractable aluminum thermode within a plexiglass casing, under precise temperature and timing control through custom-made software in MATLAB^®^ v2021b (The MathWorks Inc. Natick, MA, USA). Therefore, this device can deliver identical stimulus timing and pressure across conditions, at both painful and non-painful calibrated temperatures, allowing for an accurate comparison of neural responses in the brainstem and spinal cord.

Participants were trained to rate their pain based on two 100-point scales and to think of their rating as they felt each thermal stimulus [14,15,16]. The first scale was used to quantify pain intensity (i.e., more of the discriminative aspect of pain): 20 represents the threshold for pain, 50 is used to describe moderate pain, and 100 indicates intolerable pain. The second scale was used to quantify pain unpleasantness (i.e., more of the affective aspect of pain) and it scaled similarly: 20 represents the threshold for unpleasantness and 50 was used to describe the feeling of moderate unpleasantness, and so on. Participants were also told they could rate in increments of 5 (e.g., 25) if they felt this was a more accurate representation of the sensation they felt. Participants were asked to remember their highest pain intensity and unpleasantness ratings during the stimulation paradigm (described below) and verbally report them at the end of each trial. The stimulation device was calibrated for each participant to evoke a moderate level of pain for the Pain condition, with a limit of 52 °C to avoid peripheral tissue damage. In contrast, a constant temperature of 40 °C was used for the Sensation condition to evoke a non-painful, warm sensation. Participants were blinded to the specific temperatures used in the study and about two minutes of rest were allowed between each set of contacts in order to avoid sensitization of the skin receptors.

Following training with the stimulus and pain rating scales, participants underwent a practice run in the sham MRI to familiarize themselves with the environment and practice the stimulation paradigm before the actual imaging session. Participants were positioned in the sham MRI supine, with the RTS-2 under their right hand, and a mirror over their eyes to view a rear projection screen which displayed the pain rating scales. A practice run of the stimulation paradigm was conducted and each participant’s calibrated temperature was either confirmed or adjusted based on their ratings in the sham MRI. This process served to reduce the effects of anxiety and bulk motion during imaging.

### 2.3. Stimulation Paradigm

The stimulation paradigm (Figure 1) was designed to produce a temporal summation of pain while avoiding the habituation of receptors in the skin. Each run consisted of an initial period of 120 s with no thermode contact. At 60 s from the start of the run, participants were informed which stimulus they would experience via a rear-projection screen (i.e., painful or non-painful). This initial period was followed by 30 s of stimulation in which 10 brief contacts of 1.5 s duration occurred with onsets every 3 s. Participants were reminded before the start of each run to silently rate the intensity and unpleasantness of each thermal contact, while remembering only the highest rating on each scale. Another 120 s period followed stimulation, totalling 270 s (4.5 min) per run. Participants were asked to verbally report their pain intensity and unpleasantness ratings at the end of each run. A total of 10 functional runs were acquired for each participant, 5 in each condition (Noxious or Innocuous), in a randomized order.

### 2.4. Functional MRI Acquisition

FMRI data spanning the brainstem and spinal cord were acquired using a 3-tesla whole-body MRI system (Siemens Prisma; Siemens, Erlangen, Germany). Participants were positioned supine and were provided with padding and blankets as needed in order to minimize bulk motion and to ensure that they were comfortable in the MRI environment. The peripheral pulse was recorded using a sensor attached to the left index finger and participants were provided with a squeeze-ball to alarm the research team in the event of an emergency, or if they wished to cease their participation in the study. The RTS-2 was positioned on their right side, under the palm of the hand. Ear plugs were provided to dampen the noise of the MRI during imaging and a mirror positioned above the participants’ eyes provided them with a view of a rear projection screen which displayed the stimulation paradigm prompts and pain rating scales. Participants were instructed to limit their motion as much as possible during scanning.

Localizer images were acquired in three planes as a reference for slice positioning. In order to obtain optimal spatial fidelity and blood oxygenation level-dependent (BOLD) sensitivity in the brainstem and spinal cord, fMRI data were acquired using a T_2_-weighted half-Fourier single-shot fast spin-echo (HASTE) sequence. Images spanned a 3D volume extending from the first thoracic vertebra to above the thalamus. Nine contiguous sagittal slices were acquired with a repetition time (TR) of 6.75 s/volume and an echo time (TE) of 76 ms to optimize the T_2_-weighted BOLD sensitivity. A 28 cm × 21 cm field-of-view (FOV) was chosen with 1.46 × 1.46 mm^2^ in-plane resolution and 2 mm slice thickness. A total of 200 volumes were acquired in 5 runs for each study condition in each participant, and the runs for each condition were randomly interleaved.

### 2.5. FMRI Data Preprocessing

Functional imaging data were pre-processed and analyzed with custom-written “SpinalfMRI9” software written in MATLAB^®^v2021b (The MathWorks Inc. Natick, MA, USA). Image data were first co-registered to correct for any subtle movement of the body (i.e., bulk motion), and then slice-timing correction was applied. The timing of any detected motion was recorded for use in later pre-processing steps as well. The images were resized to 1 mm cubic voxels and spatially normalized to a 3D anatomical template, based on 300 healthy participants by normalizing these data sets to a combination of the PAM50 template of the spinal cord, and the MNI152 template of the brainstem/brain, as described by De Leener et al. [17]. The peripheral pulse was recorded in synchrony with each fMRI time series and was used to model cardiac-related physiological noise, and estimates of global noise were obtained from the white matter. These estimates of physiological noise as well as bulk motion parameters were fit to the data with a general linear model (GLM) and then subtracted from the data. These methods have been shown to be highly effective at removing physiological noise and motion effects [18]. Finally, the first two volumes of each time series were discarded to avoid variable T1-weighting. The remaining signal variations are expected to reflect descending modulation, peripheral input signalling, and local processing.

### 2.6. FMRI Data Analysis

Differences in neural activity and connectivity between Pain and Sensation conditions were investigated across regions of interest (ROIs) in the brainstem and spinal cord, which are known to be involved in pain and sensory processing, and have been investigated in previous studies conducted by our lab [6,9,19]. Regions were identified based on an anatomical region map that corresponds with the template described above, and which has been defined from multiple sources, as described previously [20,21,22,23,24,25,26,27,28,29,30,31] (https://identifiers.org/neurovault.collection:3145 accessed on 8 May 2023, www.med.harvard.edu/AANLIB/ accessed on 8 May 2023). The regions of interest include the hypothalamus (Hyp), thalamus (Thal), periaqueductal gray matter (PAG), the nucleus raphe magnus (NRM) and nucleus gigantocellularis (NGc), which are both within the rostral ventromedial medulla (RVM), the parabrachial nucleus (PBN), nucleus tractus solitaris (NTS), locus coeruleus (LC), dorsal reticular nucleus of the medulla (DRt), and the C6 dorsal horn of the spinal cord (C6RD). Voxels within these regions were functionally grouped by k-means clustering, dividing them into 5 clusters per region based on similar time-series properties. The clusters were determined based on all of the data and the same cluster definitions were then used for all analyses. That is, the same regions were compared for all conditions and all participants. Averaging data over clusters in this way allows for reduced statistical comparisons and increases the signal-to-noise-ratio over that of single voxels [32]. It is also known that a particular region may carry out a variety of functions; therefore, this method allows voxels to be divided based on function, to better account for this phenomenon. The same cluster definitions were used for all analyses and all participants.

### 2.7. Structural Equation Modeling (SEM)

Previous studies by our group have utilized structural equation modeling (SEM) in order to explain BOLD signal variance in one “target” region based on time-series characteristics of a set of “source” regions [5,19,32,33]. Using this method, we can describe connectivity between networks of regions while accounting for directionality by using a pre-defined model of the known neuroanatomy. Two time periods were selected for analysis in order to observe the effects of noxious and innocuous stimuli on descending regulation of pain during the expectation and experience of the stimuli. The time period before stimulation (“anticipation”) spans 45 s, while the time period that includes the 30 s stimulation period (“stimulation”) also spans 45 s; it is centred around the stimulus and does not overlap with the anticipation period.

SEM was conducted for the anticipation and stimulation periods in both Noxious and Innocuous conditions. A GLM was used to calculate linear weighting factors which represent the relative strength of each connection to a region (β-value). These were calculated using the following logic: if target region A receives inputs from two other source regions, B and C, and the BOLD signal time series in these responses are represented by S_A_, S_B_, and S_C_, then: S_A_ = β_AB_S_B_ + β_AC_S_C_ + e_A_, where e_A_ is the residual signal variation that is not explained by the fit. β-values were calculated for each network component in which a target region (e.g., S_A_) is paired with a unique combination of source regions (e.g., S_B_ and S_C_). The complete network used for this analysis is depicted in Figure 2.

In order to identify clusters that demonstrated the best fit to the measured data, every combination of clusters for each region was investigated throughout the network. The amount of variance in each target region that can be explained by the fit was calculated and expressed as an R^2^ value, and significance was estimated by converting R^2^ values to a Z-score by Fisher’s Z-transform. The significance of the fit was determined with previously established probability distributions. Significance was inferred at a family-wise error-corrected *p*-value of p_fwe_ < 0.05 (corresponding to uncorrected *p* < 5.25 × 10^−5^, to account for 950 unique comparisons: 38 connections and 25 unique combnations of clusters per connection), using a Bonferroni correction to account for the total number of unique connections that were tested.

The significance of β-values was assessed using a paired, two-tailed Student’s *t*-test in which a T-score was calculated from the mean and standard error across the group of participants in order to identify consistent values. Significance was inferred at a family-wise error rate corrected, p_fwe_ < 0.05. The correlation between β-values and pain scores from the Pain condition was then calculated in order to determine if variations in β-values could be explained by differences in pain ratings across individuals. This was done by converting the correlation, R, to a Z-score, in which the probability of that value occurring by random chance was estimated from a normal distribution. A threshold of *p* < 5.25 × 10^−5^ corresponding to a Z-threshold of 3.878 was again selected in order to account for multiple comparisons.

### 2.8. Analyses of Covariance (ANCOVA)

Analyses of covariance (ANCOVA) were used in order to demonstrate how each identified significant connection varied in relation to the study condition (Noxious or Innocuous) and individual differences in pain ratings (i.e., Condition × Pain Score). While significant β-values were used as dependent variables, the study condition and “normalized pain scores” for each individual were used as discrete and continuous independent variables, respectively. Significance was again inferred at *p* < 5.25 × 10^−5^. Normalized pain scores were used as an indication of relative pain sensitivity in each participant in order to avoid the assumption that each participant experiences pain the same way, and to reflect the need for different temperatures to elicit moderate pain across the group. These scores were calculated by taking the ratio of their average pain intensity rating during pain runs to the average temperature applied to their hand (Pain Score = pain rating (0–100)/temperature (°C)). Ratings for pain intensity in both Pain and Sensation conditions were used in this calculation accordingly. A larger normalized pain score indicates that a participant required lower temperatures to reach moderate pain, and therefore was more sensitive to painful stimulation. The analyses of covariance were conducted with custom software written in MATLAB^®^, and results were analyzed for both time periods before and during stimulation.

## 3. Results

### 3.1. SEM Analysis

SEM identified significant connectivity between a variety of brainstem and spinal cord regions in both Noxious and Innocuous conditions (Table 1). Connection strengths (β-values) which were significantly correlated with pain ratings at the group level are also shown in bold-face font. In the Noxious condition, a projection from the PAG to the PBN demonstrated significant connectivity (indicated by β) in the time period before stimulation. During the stimulation period, multiple brainstem regions demonstrated significant connectivity including PAG ⟶ LC, PBN ⟶ NTS, and PBN ⟶ NGc. During the Innocuous condition, significant connectivity was noted for PBN ⟶ NTS before stimulation, as well as three projections from the PAG during stimulation, terminating in the PBN, hypothalamus, and PAG.

A variety of connections demonstrating significant correlations with pain ratings before and during stimulation in the Innocuous condition were also revealed by the SEM analysis, and are shown in bold-face font in Table 1. Significant correlations between connectivity values and pain ratings were observed between the NRM and C6RD both before and during stimulation, in addition to the NTS ⟶ LC connection during stimulation. No significant correlations were observed during runs of the Noxious condition.

### 3.2. Group-Level Comparisons with ANCOVA

Analyses of covariance (ANCOVA) were conducted in order to analyze how connectivity values vary with the study condition and normalized pain scores (an indicator of individual pain sensitivity). Connections demonstrating significant main effects of the Condition or Pain Score, and Condition × Pain Score Interaction effects (*p* > 5.25 × 10^−5^) are shown in Table 2. Although no connections demonstrating a main effect of the study condition were identified before stimulation, two connections were observed to vary significantly with the study condition during stimulation, including DRT ⟶ C6RD and NTS ⟶ LC. Connections observed to vary with pain scores involved both descending projections from the brainstem to the spinal cord before stimulation as well as descending projections to the cord and LC during stimulation. Specifically, these included projections from the NTS and DRT to the spinal cord before stimulation as well as NTS ⟶ LC, PAG ⟶ LC, and NTS ⟶ C6RD during stimulation. Interaction effects were observed before stimulation for a variety of connections, including DRT ⟶ C6RD, NRM ⟶ C6RD, NRM ⟶ LC, Thal ⟶ PAG, Hyp ⟶ NRM, NGc ⟶ C6RD, and PAG ⟶ NGc. Additionally, connections including Thal ⟶ PAG, NGc ⟶ C6RD, and DRT ⟶ C6RD demonstrated interaction effects during the period of stimulation.

Connections for which the ANCOVA results identified significant effects are shown in Figure 3. The plots demonstrate how the strength of specific connections varied in relation to pain scores in each study condition. Before stimulation, the NTS ⟶ C6RD connection demonstrates increasing β-values with increasing pain scores in both conditions. Accordingly, the strength of this connection changes from negative to positive. This is similar to the PAG ⟶ LC connection during stimulation in which β-values are negative with lower pain scores and positive with higher pain scores. An opposite pattern is observed in the NTS ⟶ LC connection during stimulation in which β-values negatively correlated with pain scores.

Some connections demonstrated relatively constant values with increasing pain scores in the Noxious condition while increasing or decreasing β-values were observed during the Innocuous condition. For example, the NRM ⟶ LC connection before stimulation and the DRT ⟶ C6RD connection during stimulation illustrate this effect. Finally, before stimulation, the PAG ⟶ NGc connection demonstrated an opposite effect in which β-values varied from positive to negative values in the Innocuous condition and negative to positive values in the Noxious condition with increasing pain ratings.

In order to simplify these results, Figure 4 summarizes the results of ANCOVA analyses of connectivity values identified before and during stimulation periods. Before stimulation, the majority of connections descend from brainstem regions to the spinal cord, including pathways from the thalamus, through the PAG, and NGc, and from the hypothalamus, through the NRM. During stimulation, ascending projections are more prominent, including two connections with the LC as well as a connection from the spinal cord to the PAG. Connections which are demonstrated in both time periods include Thal ⟶ PAG and three connections terminating in the spinal cord from the NGc, NTS, and DRT.

## 4. Discussion

The objective of this study was to identify characteristics of network connectivity in the brainstem and spinal cord that are specific to pain, and to differentiate these from other effects such as anticipation or signalling related to innocuous sensations. Functional MRI was used to compare BOLD responses to noxious and innocuous thermal stimulation, revealing unique features of sensory networks in healthy participants which relate specifically to pain processing.

The results indicate fewer differences between Noxious and Innocuous conditions than initially expected, particularly in relation to the period of anticipation. The lack of connections demonstrating a main effect of a study condition in this time period may indicate similar effects of anticipation related to somatosensation on neural networks involved in responses to noxious and innocuous stimuli. This result is distinct from previous studies comparing a noxious stimulus and a condition with no stimulation at all [34]. For example, a previous study by our group used similar spinal fMRI techniques to reveal significant differences in functional connectivity between conditions with the anticipation of a painful stimulus and the anticipation of no stimulus. The differences between conditions in the aforementioned study were greater than those identified in the current investigation, suggesting that anticipation of noxious and innocuous stimuli results in more similar neural activity in the brainstem and spinal cord, than that evoked during the anticipation of no stimulus at all [5]. Therefore, the anticipation response may be less specific to pain as individuals are anticipating feeling stimulation in both conditions, regardless of the temperature.

Although few differences were found between conditions in the period of anticipation, functional connectivity between regions involved in the descending modulation of pain differed between study conditions in the stimulation period. Connections showing a significant main effect of the study condition may be more evident during stimulation because the perception of pain is a unique sensory experience, and there is evidence that it is regulated by descending influences differently than non-painful sensations [35,36]. Therefore, the connections demonstrating a significant effect of study condition in this time period (DRT ⟶ C6RD and NTS ⟶ LC) may play a significant role specific to the processing of pain and not innocuous sensations.

Descending regulation to the cord via the DRT and functions such as arousal, mediated by the NTS and LC, appear to be stronger and more consistent across individual participants during the experience of pain. This is supported in the literature considering the salient and alarming nature of pain and the evolutionary importance of being alert in situations of pain and potential danger [37,38,39]. Additionally, this finding aligns with evidence indicating that the DRT acts to enhance nociceptive responses, through a feedback loop with the dorsal horn of the spinal cord [22]. A pathway from the RVM to the DRT has been described to facilitate rapid responses to situations which could be life-threatening [40], thus supporting the involvement of the DRT which is active during pain.

Multiple brainstem regions identified in the current analysis have been shown to be involved in arousal and homeostatic regulation and this occurred differently between time periods. Specific connections linked to emotional/cognitive function were more prominent during stimulation, including those responsible for arousal. In the Noxious condition, the PAG ⟶ PBN connection was identified as significant at the group level before stimulation. The PAG is known as one of the main regions responsible for the modulation of nociceptive signals and the PBN has also been linked to nociceptive modulation [1,26,41,42]. However, during noxious stimulation, network connectivity shifted toward regions involved in sensory arousal and autonomic functioning, including regions such as the LC, NTS, NGC, and PBN [22]. In the Innocuous condition, the PAG contributed to significant connectivity during stimulation, as opposed to being involved in significant connectivity both before and during stimulation in the Noxious condition. This could indicate that the PAG is primed during the period of anticipation in order to prepare for the impending noxious stimuli, compared to the Innocuous condition in which no pain was expected. This is supported by other work indicating an important role of the PAG in states of autonomic arousal [43,44].

Generally, descending pathways were identified by connectivity analyses before and during stimulation while ascending pathways appear to be more prominent when participants were experiencing the stimulus. Our prior studies have demonstrated that the descending modulation of pain is a continuous process, occurring even in the absence of a noxious stimulus, confirming that modulatory pathways would be active during both time periods [6,9,45]. Therefore, connections such as C6RD ⟶ PAG and NTS ⟶ LC during stimulation most likely indicate pathways contributing to updating sensory information. Significant connectivity involving the NTS and LC during stimulation is also consistent with the fact that these regions are involved in arousal [46,47].

As a continuous process, regions involved in the descending modulation of pain also demonstrate a role in functional connectivity before stimulation, including the Hyp ⟶ RVM connection. Stimulation of the hypothalamus has been shown to elicit antinociceptive signalling at the level of the spinal cord [48,49] and signalling between the thalamus and PAG has also been implicated in pain modulation [6,9,50]. In the current study, this connection has relationships with both pain scores and the study condition before and during stimulation (Table 2). Connectivity between the hypothalamus and RVM has also been implicated in the affective-cognitive aspects of pain processing [51,52]. The correlation between functional connectivity and pain ratings suggests that descending signalling from the NRM to the spinal cord could be related to the magnitude of the perceived intensity of the sensory stimulus, possibly involving the descending modulatory functions of the RVM [53]. Additionally, signalling from the NTS ⟶ LC could indicate an influence of arousal on the perception of innocuous sensation, since the LC contributes to noradrenergic signalling [54].

It is possible that the observed differences among pain regions of the brainstem and spinal cord may also be attributed to individual differences in how pain is experienced. A number of connections varied with pain scores, reflecting individual pain sensitivity. These connections involve descending influences from the brainstem to the spinal cord both before and during stimulation, as well as descending influences to the LC and cord specifically during stimulation. Individuals who experienced more pain demonstrated stronger signalling between regions involved with nociceptive processing, shown by the significant relationship between β-values and individual pain scores. For example, selected connections which illustrate this finding include NTS ⟶ C6RD before stimulation and PAG ⟶ LC during stimulation. This finding was more prominent in the Noxious condition when compared to the Innocuous condition, indicating that individual differences in certain connections may be more influential when experiencing pain as opposed to a non-painful sensation.

Interindividual factors have been shown to contribute immensely to pain perception, including differences in functional connectivity [7,8,55,56]. A previous study, which focused on inter-individual differences in pain processing, identified the LC as a region with BOLD responses that were significantly correlated with individual pain ratings [57]. Along with the results of the current investigation, this suggests that the LC is a key brainstem region contributing to individual differences in how pain is perceived.

## 5. Conclusions

The current study investigated the fMRI responses of the brainstem and spinal cord to noxious and innocuous thermal stimulation, demonstrating differences in functional connectivity that appear to be specific to the experience of pain. The results indicate that, although most connections identified are likely involved in processing information related to both forms of somatosensation (noxious and innocuous), connectivity between specific regions exhibited variations in relation to study conditions and pain ratings. Additionally, brainstem regions responsible for the descending regulation of pain are active before and during stimulation while specific connections linked to emotional/cognitive function are more prominent during stimulation, including those responsible for arousal. The comparison of conditions using noxious and innocuous stimuli has revealed unique aspects of functional connectivity associated with pain and sensation as well as how these somatosensory experiences overlap at the level of the brainstem and spinal cord. This research provides a basis for further studies investigating the neural correlates of pain in humans.

## Figures and Tables

**Figure 1 brainsci-13-00777-f001:**
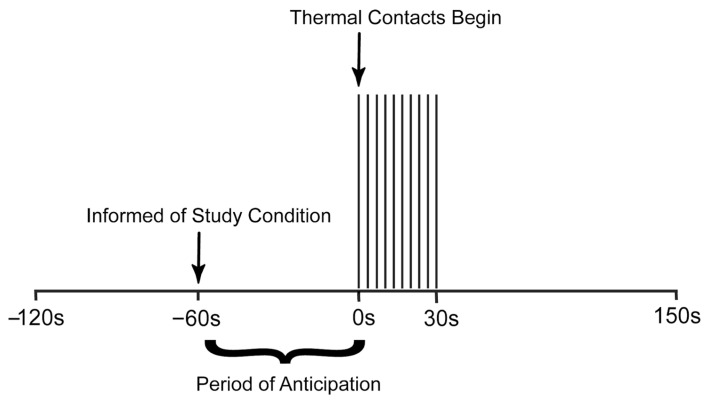
Stimulation paradigm used to elicit moderate pain and innocuous sensation in healthy individuals. The period of anticipation is shown, and time is measured in seconds relative to the onset of stimulation.

**Figure 2 brainsci-13-00777-f002:**
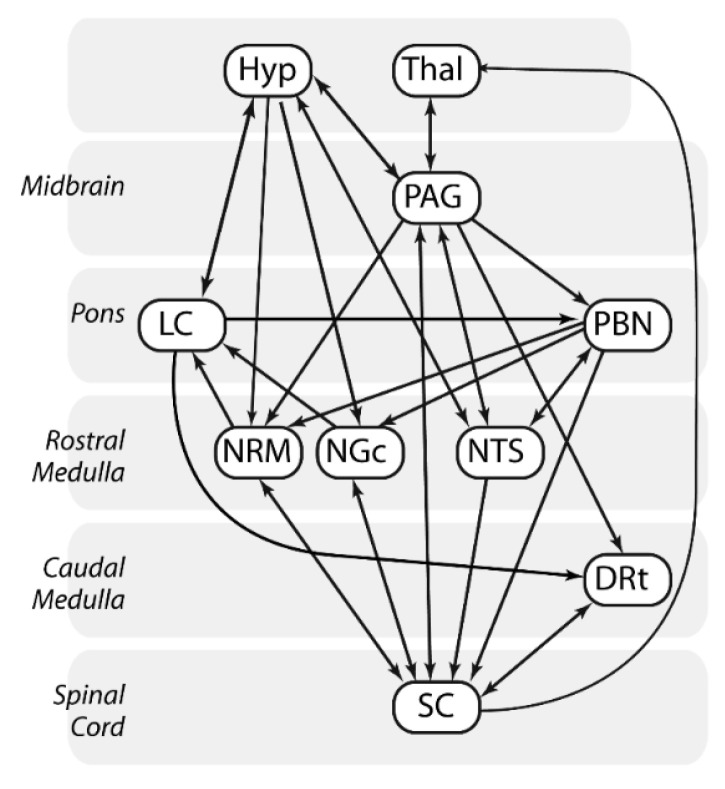
Model of connectivity between brainstem regions and the spinal cord used in SEM analyses. Connections are shown between brainstem regions including the thalamus (Thal), hypothalamus (Hyp), periaqueductal gray matter (PAG), locus coeruleus (LC), parabrachial nucleus (PBN), nucleus raphe magnus (NRM), nucleus gigantocellularis (NGc), nucleus tractus solitaris (NTS), and dorsal reticular nucleus of the medulla (DRt), as well as the C6 spinal cord dorsal horn (SC).

**Figure 3 brainsci-13-00777-f003:**
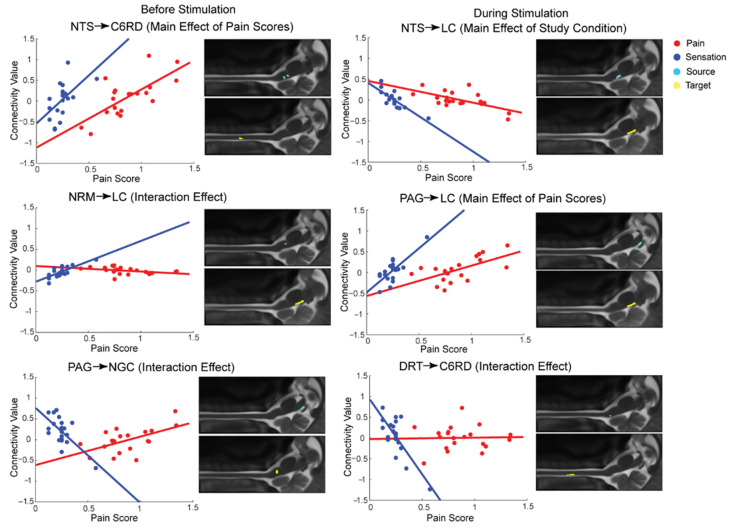
Selected neural connections demonstrating relationships between connectivity strengths (β) and pain scores in both study conditions, as identified by ANCOVA. Six connections are shown with main effects of the study condition, or pain scores, or with interaction effects. Data from Noxious and Innocuous conditions are shown in red and blue, respectively. Anatomical locations of the source and target region for each connection are shown to the right of each plot, overlaid on sagittal MR images.

**Figure 4 brainsci-13-00777-f004:**
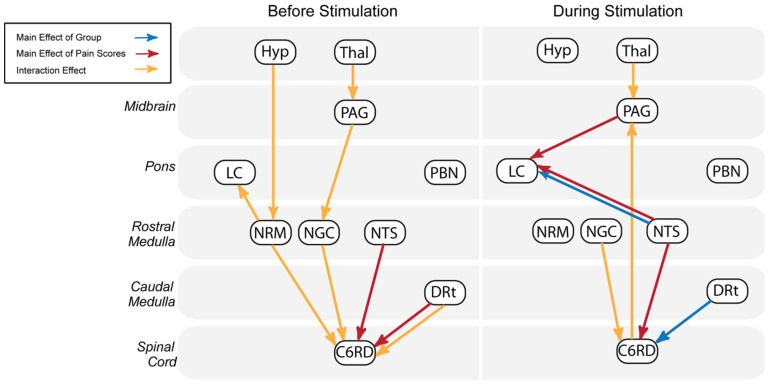
Summary of connections identified by the ANCOVA between brainstem and spinal cord regions before and during stimulation. Connections shown to significantly vary with study condition and normalized pain scores are shown by blue and red arrows, respectively. Connections displaying significant interaction effects are indicated in yellow.

**Table 1 brainsci-13-00777-t001:** Connectivity results from the SEM analysis across brainstem and spinal cord regions in Noxious and Innocuous conditions, in the periods before and during stimulation. Significant β-values are listed as the mean ± standard error across the group of participants and are identified by the T-scores (T) (paired, two-tailed *t*-test, p_fwe_ < 0.05). Values in bold represent connections with β-values that are significantly correlated with pain ratings across the group, and are identified by the Z-scores (|Z| > 3.878).

	Before Stimulation				During Stimulation			
	Connection	β	Z	T	Connection	β	Z	T
Noxious	PAG ⟶ PBN	0.16 ± 0.02	0.354	7.392	PAG ⟶ LC	0.61 ± 0.09	0.839	6.423
					PBN ⟶ NTS	0.41 ± 0.06	0.598	6.486
					PBN ⟶ NGc	0.30 ± 0.04	−0.232	7.899
Innocuous	PBN ⟶ NTS	0.28 ± 0.04	−0.420	6.615	PAG ⟶ PBN	0.16 ± 0.02	0.996	9.534
	**NRM** ⟶ **C6RD**	**0.24 ± 0.09**	**4.843**	**2.751**	PAG ⟶ Hyp	0.15 ± 0.02	0.849	6.988
					PAG ⟶ NGc	0.23 ± 0.04	0.688	6.349
					**NRM** ⟶ **C6RD**	**0.04 ± 0.03**	**4.971**	**1.212**
					**NTS** ⟶ **LC**	**0.01 ± 0.05**	**−4.699**	**0.251**

**Table 2 brainsci-13-00777-t002:** ANCOVA results identified significant connections which varied with Condition and normalized Pain Scores before and during the stimulation period. Significant connections are inferred for *p*-values less than 5.25 × 10^−5^ (F greater than 21.09).

	Before Stimulation			During Stimulation		
	Connection	*p*-Value	F-Value	Connection	*p*-Value	F-Value
Main Effect of Study Condition				DRT ⟶ C6RD	2.24 × 10^−5^	23.68
				NTS ⟶ LC	3.09 × 10^−5^	22.67
Main Effect of Pain Scores	NTS ⟶ C6RD	1.48 × 10^−5^	25.04	NTS ⟶ LC	3.24 × 10^−6^	30.25
	DRT ⟶ C6RD	2.69 × 10^−5^	23.14	PAG ⟶ LC	1.18 × 10^−5^	25.77
				NTS ⟶ C6RD	4.47 × 10^−5^	21.54
Interaction Effect	DRT ⟶ C6RD	6.46 × 10^−7^	36.27	Tha ⟶ PAG	6.92 × 10^−6^	27.62
	NRM ⟶ C6RD	1.23 × 10^−6^	33.85	NGC ⟶ C6RD	1.45 × 10^−5^	25.11
	NRM ⟶ LC	2.29 × 10^−6^	31.47	DRT ⟶ C6RD	2.29 × 10^−5^	23.66
	Tha ⟶ PAG	7.08 × 10^−6^	27.52			
	Hyp ⟶ NRM	3.02 × 10^−5^	22.79			
	NGC ⟶ C6RD	3.47 × 10^−5^	22.33			
	PAG ⟶ NGC	4.47 × 10^−5^	21.56			

## Data Availability

The data presented in this study are available on request from the corresponding author. The data are not publicly available as of yet.

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
