# Peer review of "A Comparison of Functional Connectivity in the Human Brainstem and Spinal Cord Associated with Noxious and Innocuous Thermal Stimulation Identified by Means of Functional MRI"

_brainsci, 2023, doi:10.3390/brainsci13050777_

Round 1
Reviewer 1 Report
Really well done research. Functional connectivity studies are always open to interpretation, individual variations and context. The investigators have controlled for as much of this as reasonably possible and noted the limitations adequately. This publication elegantly and substantially adds to the knowledge of descending pathways modulation of nociception while also raising new questions and providing a framework for future study.
The human experience of pain is a complex biopsychosocial phenomenon that is not well understood. Clearly the experience is different during perceived painful conditions than with innocuous stimulation of the same peripheral somatosensory receptors. But is this merely a grading of the intensity of input or are there different functional connections involved in this sorting and processing? This research seeks to understand functional connectivity in the brainstem and spinal cord during the experience of pain. The authors utilize fMRI correlated with patient reported assessments of pain to study functional connectivity in two different conditions: noxious vs innocuous stimulation. The authors conclude that “Although few differences were found between conditions in the period of anticipation, functional connectivity between regions involved in the descending modulation of pain differed between study conditions in the stimulation period.” This supports the possibility that these descending modulatory structures are involved in the “emotional-cognitive” responses to the pain experience.
Functional connectivity studies are always open to interpretation, individual variations and context. The investigators have controlled for as much of this as reasonably possible and adequately noted the relevant limitations. This publication elegantly and substantially adds to the knowledge of descending pathways modulation of nociception for example that “anticipation of noxious and innocuous stimuli results in more similar neural activity in the brainstem and spinal cord, than that evoked during the anticipation of no stimulus at all” as was seen in previous work while also raising new questions and providing a framework for future study.
Author Response
We thank the reviewer for their kind comments and for the time they took to review our paper.
Reviewer 2 Report
This study aims to investigate neural connectivity be-tween key regions involved in descending modulation of pain in response to a hot, noxious stimulus as compared to a warm, innocuous stimulus. The goal is interesting itself. However, the analysis details are not clearly stated and it is difficult to assess the validity of the results.
I have three major concerns about the methodology of the paper.
1. It seems that you assumed that the model in Figure 2 is correct. But I do not see any justification on the model. You can test whether the model is a significant one using your data, right? If the model is not significant, then your methodology lost the basis. If others uses a slightly different model, they may get totally different results.
2. The SEM model is listed in Figure 2. You stated that the model has 38 connections and 25 unique combinations of clusters per connection and therefore you have 950 unique comparisons. It is not clear why you have 950 unique comparisons. I would think that you have a huge number of unique comparisons. You have 10 nodes in Figure 2 and each node has 5 clusters. I think that you have 5^10 different models. Your multiple-comparison corrections are based on 950 comparisons. Please clarify why you have 950 unique comparisons.
3. You have 5 clusters for each region based on k-means clustering algorithms. So for each subject and each condition, the 5 clusters are at different locations. Once a significant correlation is found with any of 25 combinations, it was claimed as a significant correlation. I have a concern of using different locations across different subjects and conditions.
A few other (mostly minor) concerns are listed below.
4. Title has a spelling error “spinal sord” -> “spinal cord”.
5. Pg. 4, the 7th to last line. “MN152” -> “MNI152”
6. Pg. 4, the 3rd to last line. Have you recorded respiratory noises? What you are referring to as a bulk motion? Global noise from the white matter? or motion correction parameters?
7. Pg. 5, FMRI Data Analysis, line 10, What does both refer to as?
8. Pg. 5, FMRI Data Analysis, line 13, see my comments #3. you may have different clusters from different subjects and conditions.
9. Pg. 5, Structural Equation Modeling (SEM), lines 8-9, why 30-second stimulation period spans 45 seconds?
10. Pg. 6, 950 unique comparisons, see my comments #2.
11. Pg. 7, SEM Analysis, line 1. see my comments #1. Is your SEM model a significant model? If yes, you should report the significance of the model. If not, you should not report the significant connectivity, right?
12. Pg. 7, Table 1. You have “Sens” in the Table 1. It seems that you talked about the Innocuous condition in the text. Please be consistent with your terminology.
13. Pg. 7, the 4th to last line. For the bold-face connections, were they shown as significant connections? It seems not from your presentation. How can we interpret that these connections are not significant, but are significantly correlated with pain ratings?
Author Response
We have repeated the reviewer's comments for clarity:
I have three major concerns about the methodology of the paper.
- It seems that you assumed that the model in Figure 2 is correct. But I do not see any justification on the model. You can test whether the model is a significant one using your data, right? If the model is not significant, then your methodology lost the basis. If others uses a slightly different model, they may get totally different results.
Response: Thank you for this comment. The model is based on detailed descriptions of descending pain regulation networks by Millan (Millan 2002, reference 22) as described in the Methods section. Any model of connectivity between regions used for SEM applied to fMRI data must be a simplification of the real, more complex, anatomy. It is not possible to model beyond the limits of the spatial resolution of the fMRI data. However, it is valid to model the general properties of the network to see how the model fits the measured BOLD data. All of the results shown are significant fits to the chosen model, and are sufficiently consistent across the group of participants, or correlated with pain ratings, to show differences between the study conditions. Although it is certainly possible to test other network models, as pointed out by the reviewer, we have shown that the chosen model fits the fMRI data.
- The SEM model is listed in Figure 2. You stated that the model has 38 connections and 25 unique combinations of clusters per connection and therefore you have 950 unique comparisons. It is not clear why you have 950 unique comparisons. I would think that you have a huge number of unique comparisons. You have 10 nodes in Figure 2 and each node has 5 clusters. I think that you have 5^10 different models. Your multiple-comparison corrections are based on 950 comparisons. Please clarify why you have 950 unique comparisons.
Response: Thank you for this comment. We describe in the Methods section how the BOLD responses in each “target” region are fit to the measured BOLD responses in the “source” regions, as specified in our assumed network model. Each target region is fit one at a time. That is, we do not need to repeat the fitting procedure to a region/cluster with every combination of clusters in another region to which it is not connected. With 5 clusters in the target region, and 5 clusters in a source region, this makes 25 combinations. However, each target has multiple source regions. If the fit is repeated multiple times with the same cluster in the target, the same cluster in source 1, and multiple combinations of clusters in other sources, then these cannot be considered to be independent comparisons. The results will be related because the same data are used for the target cluster, and for one of the source clusters. In order to correct for multiple comparisons, it is necessary to correct for the number of independent statistical tests. We have determined in previous studies (cited in the paper) that a suitable correction for multiple comparisons is achieved by accounting for the fact that each connection in the model has 25 unique combinations of clusters (when we have 5 clusters in each region). With 38 connections this therefore gives 950 independent statistical comparisons.
- You have 5 clusters for each region based on k-means clustering algorithms. So for each subject and each condition, the 5 clusters are at different locations. Once a significant correlation is found with any of 25 combinations, it was claimed as a significant correlation. I have a concern of using different locations across different subjects and conditions.
Response: Thank you for your comment. As described in the Methods, the clusters were determined based on all of the data, and then the same cluster definitions were used for all analyses, for each participant. The clusters are not at different locations for different participants or for different study conditions. A connection is inferred to be significant if the average b value across the group was significantly different than zero, or was significantly correlated with pain ratings.
We have clarified this by adding text to the paragraph on page 5:
“The clusters were determined based on all of the data and the same cluster definitions were then used for all analyses. That is, the same regions were compared for all conditions and all participants.”
Minor points
A few other (mostly minor) concerns are listed below.
- Title has a spelling error “spinal sord” -> “spinal cord”.
Response: This has been corrected in the text
- Pg. 4, the 7th to last line. “MN152” -> “MNI152”
Response: This has been corrected in the text
- Pg. 4, the 3rd to last line. Have you recorded respiratory noises? What you are referring to as a bulk motion? Global noise from the white matter? or motion correction parameters?
We mention earlier in the same paragraph that we corrected the data for bulk motion, and then later in the paragraph we describe how we also regress out any residual effects of bulk motion from the data. Bulk motion refers to any movement of the body, not cardiac or respiratory motion. We have revised the paragraph to make this more clear with the following sentences:
“Image data were first co-registered to correct for any subtle movement of the body (i.e. bulk motion), and then slice-timing correction was applied. The timing of any detected motion was recorded for use in later pre-processing steps as well.”
- Pg. 5, FMRI Data Analysis, line 10, What does both refer to as?
“Both” refers to the nucleus raphe magnus (NRM) and nucleus gigantocellularis (NGc) which are mentioned just prior to that point in the same sentence. We have made the sentence more clear by inserting the word “the” and removing a comma.
- Pg. 5, FMRI Data Analysis, line 13, see my comments #3. you may have different clusters from different subjects and conditions.
This point has been addressed above, by adding text to make this point more clear.
- Pg. 5, Structural Equation Modeling (SEM), lines 8-9, why 30-second stimulation period spans 45 seconds?
It doesn’t. We have revised the sentence to read:
“The time period before stimulation (“anticipation”) spans 45 seconds, while the time period that includes the 30-second stimulation period (“stimulation”) also spans 45 seconds, it is centred around the stimulus and does not overlap with the anticipation period.”
- Pg. 6, 950 unique comparisons, see my comments #2.
This point has been addressed above.
- Pg. 7, SEM Analysis, line 1. see my comments #1. Is your SEM model a significant model? If yes, you should report the significance of the model. If not, you should not report the significant connectivity, right?
This point has been addressed above.
- Pg. 7, Table 1. You have “Sens” in the Table 1. It seems that you talked about the Innocuous condition in the text. Please be consistent with your terminology.
The row headings in the table have been corrected.
- Pg. 7, the 4th to last line. For the bold-face connections, were they shown as significant connections? It seems not from your presentation. How can we interpret that these connections are not significant, but are significantly correlated with pain ratings?
The significance of our results was inferred in one of two ways in every case. Either the average b value for a group could be significantly different than zero, and therefore indicates a consistent effect, or the b values could be correlated with pain ratings across participants in the group. Both types of results demonstrate a significant effect. If b values are significantly correlated with pain ratings then it stands to reason that the b values vary across participants and are not likely to be significantly different than zero. That is, the same b values are unlikely to be simultaneously significantly different than zero and also correlated with pain ratings. The caption to Table 1 explains that the bold-face font indicates the connections with significant correlations between b values and pain ratings, with Z-scores with magnitudes greater than 3.878. However, we have corrected the error in the caption and revised the text to read (|Z| > 3.878)
Reviewer 3 Report
Koning et al. following up on the previously published work by the same group (PMID: 35599729) and others (PMID: 34697093, 35080494, 31579846) investigated the functional connectivity in the brainstem and spinal cord upon different thermal stimulation in human subjects. Due to the similarity to the previous published works, the novelty is low. The presentation of the data and the organization of the manuscript can be improved. My itemized suggestions are listed below:
1. Typo in the title: spinal “sord”. I believe the authors meant spinal cord.
2. Fig.1 is listed to describe a method used. It is unusual to designate a figure for the Materials and Methods.
3. Fig. 2 is not mentioned in text.
4. Table 2:
a. Why there are two p values? If the authors aim to compare Before and During Stimulation, there should be one p value per comparison. Please clarify the comparison for each p value.
b. Please include how F value is calculated and what F critical value is set.
5. Fig. 3: Please enlarge the images to allow make the signals more visible.
6. I am surprised that the authors did not cite the previous work from the same group which is highly relevant, PMID: 35599729.
Author Response
We have repeated the reviewer's comments for clarity:
Koning et al. following up on the previously published work by the same group (PMID: 35599729) and others (PMID: 34697093, 35080494, 31579846) investigated the functional connectivity in the brainstem and spinal cord upon different thermal stimulation in human subjects. Due to the similarity to the previous published works, the novelty is low. The presentation of the data and the organization of the manuscript can be improved. My itemized suggestions are listed below:
Response: Respectfully, we find the reviewer’s comments to be somewhat harsh given the minor nature of the concerns listed below. Although the methods used in this paper are not novel, since we have used them before, the study of differences in neural responses to innocuous and noxious thermal stimuli is novel and little previous published information is available on this topic. Our findings are somewhat unexpected and are certainly novel. The purpose of this paper was not to present new analysis methods.
- Typo in the title: spinal “sord”. I believe the authors meant spinal cord.
Response: Thank you for pointing out this error, we have fixed it in the revised version of the manuscript.
- 1 is listed to describe a method used. It is unusual to designate a figure for the Materials and Methods.
Response: Respectfully, it is not at all unusual to include a figure that is necessary to explain the methods.
- 2 is not mentioned in text.
Response: Figure 2 is mentioned in the text in the sentence directly before the figure appears.
- Table 2:
- Why there are two p values? If the authors aim to compare Before and During Stimulation, there should be one p value per comparison. Please clarify the comparison for each p value.
Response: The results are not comparing the time period Before Stimulation to the time period During Stimulation. The results are from ANCOVAs identifying how the results vary with both the study conditions (noxious and innocuous stimulation) and the pain ratings. This analysis was done independently using data from a 45 second time period before stimulation, and again using data from a 45 second period spanning the 30 seconds of stimulation.
- Please include how F value is calculated and what F critical value is set.
Response: The F value is the statistic calculated using the ANCOVA. It is beyond the scope of this manuscript to explain well-established statistical methods.
- 3: Please enlarge the images to allow make the signals more visible.
Response: We assume that the reviewer is referring to the visibility of the regions shown overlying the anatomical images. It is necessary to show all of the anatomical images at the same size and spanning the same extent of the anatomy in order to demonstrate the different anatomical locations of different regions/clusters. We feel that enlarging the anatomical images relative to the size of the plotted data points (connectivity values vs pain scores) would take the focus off of the plots and that this would obscure the main purpose of this figure. The reader always has the option of displaying the figure larger on their computer screen.
- I am surprised that the authors did not cite the previous work from the same group which is highly relevant, PMID: 35599729.
Response: The paper mentioned by the reviewer presents a study of altered neural signaling in fibromyalgia. This is not the same topic as the present paper which investigates differences in responses between innocuous and noxious stimuli. Our group has published many other papers using connectivity analysis with structural equation modeling to investigate pain responses. It is not clear why the reviewer would single out this specific paper, and would be surprised that we did not cite it.
Round 2
Reviewer 2 Report
The presentation of the paper have improved from the previous version. However, I still have a few questions, which require further clarification.
Major points
- It seems that you assumed that the model in Figure 2 is correct. But I do not see any justification on the model. You can test whether the model is a significant one using your data, right? If the model is not significant, then your methodology lost the basis. If others uses a slightly different model, they may get totally different results.
Response: Thank you for this comment. The model is based on detailed descriptions of descending pain regulation networks by Millan (Millan 2002, reference 22) as described in the Methods section. Any model of connectivity between regions used for SEM applied to fMRI data must be a simplification of the real, more complex, anatomy. It is not possible to model beyond the limits of the spatial resolution of the fMRI data. However, it is valid to model the general properties of the network to see how the model fits the measured BOLD data. All of the results shown are significant fits to the chosen model, and are sufficiently consistent across the group of participants, or correlated with pain ratings, to show differences between the study conditions. Although it is certainly possible to test other network models, as pointed out by the reviewer, we have shown that the chosen model fits the fMRI data.
Have you tested that the entire model in Figure 2 (including all the nodes and connections) is a significant model? I do not see that you reported the significance of the model.
I do not see that you referenced 22 anywhere in your text either although that reference appears in your reference list.
From your response to point #2, it seems that you built a model for each “target” region. If that is the case, can you clearly state that you had many “smaller” models and report the significance of each “smaller” model (for instance, a smaller model Hyp->NGC and PBN ->NGC)? In my opinion, only if it is a significant “smaller” model, you can do a further analysis for the beta values.
- The SEM model is listed in Figure 2. You stated that the model has 38 connections and 25 unique combinations of clusters per connection and therefore you have 950 unique comparisons. It is not clear why you have 950 unique comparisons. I would think that you have a huge number of unique comparisons. You have 10 nodes in Figure 2 and each node has 5 clusters. I think that you have 5^10 different models. Your multiple-comparison corrections are based on 950 comparisons. Please clarify why you have 950 unique comparisons.
Response: Thank you for this comment. We describe in the Methods section how the BOLD responses in each “target” region are fit to the measured BOLD responses in the “source” regions, as specified in our assumed network model. Each target region is fit one at a time. That is, we do not need to repeat the fitting procedure to a region/cluster with every combination of clusters in another region to which it is not connected. With 5 clusters in the target region, and 5 clusters in a source region, this makes 25 combinations. However, each target has multiple source regions. If the fit is repeated multiple times with the same cluster in the target, the same cluster in source 1, and multiple combinations of clusters in other sources, then these cannot be considered to be independent comparisons. The results will be related because the same data are used for the target cluster, and for one of the source clusters. In order to correct for multiple comparisons, it is necessary to correct for the number of independent statistical tests. We have determined in previous studies (cited in the paper) that a suitable correction for multiple comparisons is achieved by accounting for the fact that each connection in the model has 25 unique combinations of clusters (when we have 5 clusters in each region). With 38 connections this therefore gives 950 independent statistical comparisons.
The logic is still not clear here. This is worth further explanation. Let us talk about a simple model. You have three nodes Hyp, NGC, and PBN with the connections Hyp->NGC and PBN ->NGC. Each of three nodes have 5 clusters, let us say, Hyp has clusters 1, 2, 3, 4, and 5. In this simple model, each of Hyp, NGC, and PBN has 5 possible clusters. Do you have totally 5^3 possible models? Why should the number of possible models be 5^2 * 2?
Minor points
A few other (mostly minor) concerns are listed below.
- Pg. 5, Structural Equation Modeling (SEM), lines 8-9, why 30-second stimulation period spans 45 seconds?
It doesn’t. We have revised the sentence to read:
“The time period before stimulation (“anticipation”) spans 45 seconds, while the time period that includes the 30-second stimulation period (“stimulation”) also spans 45 seconds, it is centred around the stimulus and does not overlap with the anticipation period.”
Let us look at your Figure 1. Period of anticipation is from -60s to 0s. Stimulation period is from 0s to 30s. How do the 45 seconds of simulation period not include anticipation with it center around the stimulus? Do you mean to set up the stimulation period from 0s to 45 s?
Author Response
We have repeated the authors comments for the 2nd round in italics for clarity.
I do not see that you referenced 22 anywhere in your text either although that reference appears in your reference list.
1) Reference 22 is included in the Methods section:
"Regions were identified based on an anatomical region map which corresponds with the template described above, and which has been defined from multiple sources, as described previously (20-31)"
Have you tested that the entire model in Figure 2 (including all the nodes and connections) is a significant model? I do not see that you reported the significance of the model.
From your response to point #2, it seems that you built a model for each “target” region. If that is the case, can you clearly state that you had many “smaller” models and report the significance of each “smaller” model (for instance, a smaller model Hyp->NGC and PBN ->NGC)? In my opinion, only if it is a significant “smaller” model, you can do a further analysis for the beta values.
2) The reviewer seems to be thinking of this as a connectivity analysis using correlation, which is described in terms of nodes and edges. The SEM method that we present can be more accurately thought of as a linear regression between BOLD responses in multiple regions in order to identify relationships between them. The “significance of the model” can only be described in terms of how well it explains the observed BOLD responses. The model has been validated based on published studies of the neuroanatomy, and with prior SEM studies by our group.
3) The significance of the fit is determined for each “smaller” network as the reviewer points out. We do not determine the significance for the total combined network, since each target region is fit, one at a time. It is more accurate to think of the results as representing multiple connected smaller networks. This is described in the Methods section.
The logic is still not clear here. This is worth further explanation. Let us talk about a simple model. You have three nodes Hyp, NGC, and PBN with the connections Hyp->NGC and PBN ->NGC. Each of three nodes have 5 clusters, let us say, Hyp has clusters 1, 2, 3, 4, and 5. In this simple model, each of Hyp, NGC, and PBN has 5 possible clusters. Do you have totally 5^3 possible models? Why should the number of possible models be 5^2 * 2?
4) A correction for multiple comparisons is to correct for independent statistical comparisons. It would be an over-correction to count each combination of clusters as an independent test. The number of independent tests in the example provided by the reviewer is 5^2 * 2. This because for cluster 1 of the Hyp, we test all combinations of 5 clusters for NGC and PBN. Then, for cluster 2 of the Hyp, we test the same combinations of 5 clusters for NGC and PBN. A significant connection between the NGC and PBN that appears for one combination of clusters by random chance (due to random noise), would likely occur for all choices of clusters for the Hyp. These are not independent statistical tests. Our previous studies to test the significance thresholds and model the false positive rate with null data (simulated from normally distributed random numbers) support this method of choosing our statistical threshold and correcting for multiple comparisons (Stroman et al., 2016).
Let us look at your Figure 1. Period of anticipation is from -60s to 0s. Stimulation period is from 0s to 30s. How do the 45 seconds of simulation period not include anticipation with it center around the stimulus? Do you mean to set up the stimulation period from 0s to 45 s?
5) Yes, we state that the 45 second period spans the stimulation period. This means that the 45 second period includes the stimulation period, and clearly it is longer than the 30 second stimulation period, so it includes some time before and after stimulation. The ability to see variations in BOLD responses due to the stimulus, and how these are related across regions, is facilitated by including time points before and after stimulation.
Reviewer 3 Report
1. Table 2:
b. Please include what F critical value is set.
Author Response
The F-critical value that corresponds to our stated threshold of p < 5.25 x 10-5, is 21.09. This has been added to the caption for Table 2.